# Empirical Study on Design Trend of Taiwan (1960s–2020): The Evolution of Theme, Diversity and Sustainability

**Jianping Huang** [1] , **Yuheng Tao** [2], **Minghong Shi** [2,3,*] **and Jun Wu** [4]

1   School of Fine and Arts, Fujian Normal University, Fuzhou 350117, China
2   Graduate School of Creative Industry Design, National Taiwan University of Arts,
    New Taipei City 22058, Taiwan
3   College of Creative Design, Shenzhen Technology University, Shenzhen 518118, China
4   Department of Digital Media Art, School of Arts and Design, Shenzhen University, Shenzhen 518061, China
*   Correspondence: shiminghong@sztu.edu.cn

**Abstract:** With significance in improving and developing local design culture as well as in supplementing global design history, this essay describes a study on the past and a clear prediction of the future by exploring Taiwan's design history from approximately the 1960s to 2020 based on the evolution of theme, diversity, and sustainability. In this research, the Python programming language is used to apply three algorithms of term frequency–inverse document frequency (TF-IDF), Simpson's diversity index (SDI), and latent Dirichlet allocation (LDA) to conduct a text exploration of design journals. The results show the following: in the 1960s–1980s, the evolution of theme focused on evaluation strategies, technical practices, and foreign cultures, on digital design, multiculturalism, and design aesthetics in the 1990s, and on emotional human factors, intelligent technology, and local culture since the beginning of the 21st century. Local culture and intelligent technology are the main driving forces of the current design industry. Regarding diversity, after a period of rapid change and stable rising, it has shown a downward trend in recent years. This indicates that current design needs to be stimulated by external environmental variations. Sustainability was focused on technology, the market, and education during the 1960s–1980s; on consumers, design education, and eco-design during the 1990s; and on integration across fields during the 2000s–2020. In order to gain a wider perspective of the complete design context of Chinese culture, the results show the current and future trends of the academic community, in addition to a reference for the study of the design histories of other areas in the world.

**Keywords:** design context; text mining; periodicals; trending topics; trend of diversity; sustainable design; glocalization

## 1. Introduction

With the change in global design in the 1960s, modern design rapidly transformed into postmodern design [1]. At that time, both industrial and academic circles questioned the modernism that had emerged in the early 20th century, as well as the functionalist and rationalist design concepts upheld by the modernists, such as "Form Follows Function", which was thought to be the main reason for the failure of modern urban development [2]. Such questioning and resistance gradually resulted in the formation of another independent value that has divided the cultural field, a sort of intellectual irrationalism, moral cynicism, and emotional hedonism, and a form-limited principle that rejects modernism. However, Zhou (2018) pointed out that it was essentially the continuation and development of modernism, namely, a postmodernist period characterized by "diversity" [3]. In such a context of inheritance and transition, it can be seen that the 1960s represented a critical transition period for modern design, representing the transition from rational design to emotional design, as well as that from modernism to postmodernism.

In globalization and culture, while enthusiasm dominated as a viewpoint in the 1990s regarding the notion of globalization, in the aftermath of the 2008 Great Recession, the worsening of the COVID-19 pandemic, and accelerated cultural conflict among nations and ethnic groups, skepticism around globalization has become widespread [4]. Hence, the notions of globalization and glocalization are a means for grasping an increasingly complicated reality [4]. Studies have pointed out that the problematic nature of global–local relations has emerged as a key research theme [4–6]. In the era of integration and globalization, it not only bears significance in the study of local development pathways but also created the development modes of local cultures. In addition, it also presents a more diversified global culture, which can subsequently alleviate the conflicts and promote global cultural integration and development.

With the greater attraction to globalization, more topics concerning globalization have emerged, e.g., the disposition of global strategy in the issue of climate change [7]. The COVID-19 pandemic has increased the importance of global strategies for sustainable development [8,9]. 'The Hannover Principles' noted: sustainable globalization attributes give rise to mutual dependence among the human system, industry, and the environment [10]. Therefore, the question of how to achieve sustainable development in resource-based areas is also a concern of governments [11–13]. However, design serves a key roles in sustainability [14,15]. Since the 1960s, sustainable design can be considered as the discussion of 'the ethics of design' from McDonough's green economy concept of 'Cradle to Cradle: Remaking the Way We Make Things' and 'Green Design, Universal Design, Friendly Environment' in a broad sense [10,16]. Since the rapid industrialization of many nations in Asia, the consequent rapidly rising levels of water, air, and land pollution have raised concerns about the unsustainability of current growth patterns [17]. It is important to explore the development pathway of sustainable design in Asia and to provide a reference for other areas in the world to strengthen their knowledge of environmental obligations in a regional and even global sense.

The design of Asia has acquired a key role [18,19]. However, design cultures are different across Asia and are interconnected [20]. In the era of globalization, how do designers find the balance between local culture and international ideological trends, and form their own unique expression of visual language? How will the design context of Chinese culture act as a key component of Asian culture development in the future? However, the current construction of modern design history is mainly focused on recording the development of Western modern design, while Chinese culture and even all Asian regions are almost in a subordinate position [21]. The history of design, as a discipline, was based on assumptions on what design is and how we studied design in the past, and we should not ignore the dynamic crossing of regional knowledge boundaries. Thus, the current design researches should have included new subjects, such as design researches involving Asia, Africa and other regions [22]. Current research into the history of modern design against the background of Eastern culture is still limited, and it is not enough to constitute a complete modern design history with regional cultural characteristics. Therefore, with the increasingly serious trend of global cultural homogeneity, it is of positive significance to reinforce design history research from the perspective of Eastern culture by organizing the development context of Eastern history and culture, and by exploring the relations between social background and design development.

As one of the centers of Chinese cultural heritage, Taiwan also existed in an important period for the initial development of modern design in the 1960s [21,23]. Since the 1960s, Taiwan's design industry has undergone original equipment manufacturer (OEM), original design manufacturer (ODM), and original brand manufacturer (OBM) processing. These three stages are not only a sign of the times but also represent the transformation of design thinking, the development of creative tools, and the evolution of technical styles [24]. Taiwan's cultural variety and distinction offer a potential application in the field of design; especially because designing local features into products appears to be increasingly important for the global market, cultural features are considered to be unique characteristics

that can be embedded into a product for both the enhancement of its identity in the global market and for a better individual consumer experience [25]. Furthermore, as Taiwan is typically a resource-limited area, experts have noted that exploring Taiwan's sustainable development trend bears indicative significance for globalization [26–28].

With the development of the discipline, the literature has become an important source of information [29]. In the period of underdeveloped information, "periodicals", as one of the main methods of information dissemination, became an important part of the literature. The systematic analysis of the literature published in academic journals was not only conductive to the tracing of the latest achievements of the academic community, and to updating and enriching the existing knowledge system, but also provided a reference to the theories and practices of teaching. It also helped predict the development and evolution of related disciplines, future directions, and thematic trends [30–32].

In summary, this study adopted text mining to study the development context of Taiwan's design industry from the 1960s to 2020 as well as to study the evolution of theme, diversity, and sustainable design trends using the Python language. We adopted the following three research objectives:

- To study the evolution of theme concerning Taiwan's design development from the 1960s to 2020;
- To study the diversity of Taiwan's design development from the 1960s to 2020;
- To explore the sustainability of Taiwan's design development from the 1960s to 2020.

The remainder of this paper is organized as follows: Section 2 briefly reviews the development of Taiwan's modern design from the 1960s to 2020. Section 3 describes the text-mining method in the Python programming language, research materials, and processes. Section 4 introduces the empirical results of the evolution of theme, diversity, and sustainability. Section 5 discusses the empirical results of these three aspects. Finally, Section 6 presents the conclusion.

## 2. Taiwan's Design Context

A national design's development process relies on the planning and support of strong and relevant policies [33]. Raulik-Murphy et al. (2010) put forward the concept of a national design system based on the national innovation system, economic cooperation, and development, namely, the design development framework of a country. They pointed out that a national design system should contain four elements: design support, design promotion, design education, and design policy [34]. In addition, Wang (2016) also pointed out that the academic theories are derived from the rational thinking of social intellectuals on artistic design and are constructed based on social, political, economic, and cultural advancements [35]. Hence, this paper combines the development of four aspects (design policy, design education, social activity, and academic research) of Taiwan's modern design from three periods (1960s–1970s, 1980s–1990s, and 2000s–2020), as shown in Table 1.

In summary, through the above-mentioned analysis of design development in Taiwan from the 1960s until today, we can roughly understand modern design development in Taiwan: in the OEM stage from the 1960s to the 1970s, Taiwan started a dialogue with global design through a two-way exchange strategy including "import" and "export", resulting in the rapid development of domestic modern design; during the ODM stage from the 1980s to the 1990s, Taiwan promoted a cross-domain cooperation strategy across "human factors" and "digital" as well as practiced user-friendly design through "technology based on people"; during the OBM stage, which started at the beginning of the 21st century, Taiwan advocated for the direction of aesthetics and cultural creation that integrates "emotion" and "technology" and promoted "emotional" design, which integrates technology into design and creativity.

**Table 1.** A list of the literature on research into trends in the field (organized in this study).

| | |
|---|---|
| **First Stage: 1960s–1970s, The Stage of Technological Development Centered on Production** | |
| Design policy | The Industrial Development Bureau of the Ministry of Economic Affairs aimed to guide national design development strategies in response to changing industrial needs [33]. In 1967, the Chinese Industrial Design Association (CIDA) was founded, and the China Industrial Design and Packaging Center was founded in 1973, to assist in the transformation of Taiwan's economic and industrial structure. |
| Design education | Early design education gradually took shape in the fine arts discipline. In 1953, the Art Department of the Normal School began to teach courses in pattern design and color science. Up to 1957, the field of design education gradually shifted from pure arts to arts and crafts. In 1962, the National Taiwan University of Arts restructured the art printing and art engineering specialties to three-year majors, officially opening the door to design education [36]. Additionally, foreign experts were invited to teach in Taiwan [37]. |
| Social activity | The earliest design concept in Taiwan came from the Council on U.S. Aid (CUSA), which promoted industrialization in Taiwan through policies such as introducing foreign scholars and experts, as well as using public funds to send talents to study abroad [38]. In 1966, the China Productivity and Trade Center sent some chosen students to study in Japan, which played an important role in design development in Taiwan [39]. At the same time, private institutions also prepared to set up organizations to promote handicrafts and the industry [40]. |
| Academic research | With the multi-dimensional promotion and popularization of the design industry due to policy strategies and social activities, a large number of publications introducing design were also issued, such as *Designer*, *Designer & Designing*, and *Industrial Design*, which brought then-current design concepts to the information-poor design industry. |
| **Second Stage: 1980s–1990s, the Stage of Cultural Connotation Centered on Design Image** | |
| Design policy | From 1989 to 2004, The Foreign Trade Association formulated a three-phase five-year plan [24]. The 1989–1994 "Industrial Design Development Strategy Five-Year Plan" and the 1995–1999 "Comprehensive Product Capability Improvement Plan" led the design industry in Taiwan from germination to maturity. Taiwan's human factors engineering was enlightened and developed during this period. In 1984, the "Human Factors Engineering Promotion Group" established by the National Science Committee for Long-term Development improved academic research and related technical standards for domestic human factors engineering and accelerated international exchanges in research related to human factors engineering [41]. |
| Design education | Industrial design education gradually received attention, and the design profession gradually became popular. University courses were basically based on the Bauhaus education system, providing functional practice-related courses. By the 1990s, numerous schools in Taiwan had decided to integrate resources and merge their art-, design-, and media-related departments for the establishment of design colleges, art colleges, and masters' research institutes. |
| Social activity | The Product Design and Packaging Center, Design Promotion Center/CETRA (DPC/CETRA), Fashion Color Association, Handicraft Development Center, and other institutions were founded during this period to guide the design ability and technology of individual key industries through multi-point public legal entities. |
| Academic research | Under the impact of the regional design trend of postmodernism, the topic of "Taiwan Studies" gradually gained attention during this period. Many monographs related to local design culture emerged during this period [42,43]. |
| **Third stage: 2000s–present, the Stage of Emotional Technology Centered on Brand Image** | |
| Design policy | The Executive Yuan promoted the "Challenging the 2008 National Development Key Plan" in 2002, and approved the "Taiwan Design Industry Take-off Plan", as policy tools to promote the design industry. Such policies helped integrate design continuously into life, provide support to the industry, cultivate talents, and enhance international competitiveness. The Legislative Yuan passed the "Cultural Creation Law" in January 2010 to further promote the use of "Cultural Creation" as "soft power" for the country's development [24]. |
| Design education | New design disciplines emerged in response to the development of the science and technology design industry. In 2001, Mingdao University created the first department, named the "Digital Design Department", with digital media as its main body, information technology as its tool, and design integration as the goal. In terms of the academic system, doctoral education was further developed in the field of design to promote academic research education. |
| Social activity | The implementation of policy plans has made the design industry in Taiwan more competitive, as Taiwan successfully gained the right to hold the Taipei World Design Expo 2011, the World Design Capital Taipei, and the World Universiade, having laid a good foundation for design development in Taiwan [44]. |
| Academic research | In 2001, Taiwan developed its knowledge economy and put forward the concept of "cultural power", which transforms culture into economic advantage by observing globalized industry development and the knowledge-based economy. With the development of intelligent technology, the further promotion of user experience and participatory design trends gave designs more diverse perspectives [45]. |

## 3. Materials and Methods

According to the research purposes, this study selected the main text contents published by *Industrial Design* since its inception in 1967 and used text-mining techniques for analysis. This section explains the research methods and steps, including data preprocessing, article categorization structure, categorization work progress, and categorization reliability calculation.

### 3.1. Research Objects: Industrial Design

*Industrial Design* has existed for more than half a century since its establishment. As the longest-lived publication among the existing design journals in Taiwan, it records the relatively complete history of design development in Taiwan [21]. Professionals have studied the cover design change, as shown in Figure 1, from points, lines, and planes to photography, and then to computer graphics, reflecting the technology development of the Taiwanese design industry [24]. The author (2019) compared the four basic visual elements from the perspective of current psychological group cognition by selecting the covers of the first 10 issues of the *Industrial Design* magazine and 10 posters from the Bauhaus period. The research results show that early designs in Taiwan mostly originated from the design ideas of the German Bauhaus blended with Japanese culture, and the modeling style developed towards a rational geometric function. These studies show that *Industrial Design* has great importance for the development and influence of design in Taiwan [21]. Therefore, the design journal was chosen as the subject in this study in order to conduct an analysis according to the research targets mentioned above.

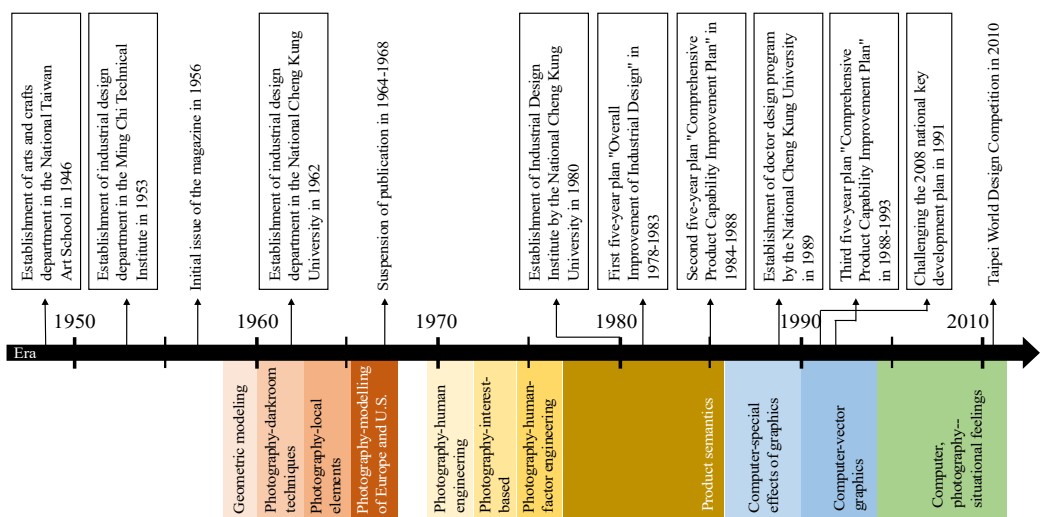

**Figure 1.** Evolution of the cover design concepts of *Industrial Design* magazines (Redraw from [24]. Copyright 2013 Lin and Lin).

### 3.2. Research Methods

As a constantly advancing technological tool, text mining is applicable to processing a huge amount of text data. It performs document editing, organization, and analysis processes by combining qualitative and quantitative research and using linguistics, statistical analysis, and other techniques. The purpose of this is to ascertain content facts and trends and mine the useful value of hidden features [46–49]. The main procedures of text mining include data retrieval and processing, word segmentation, feature selection, categorization and clustering, text representation, and interpretation [50,51]. It can be used to analyze and explore academic literature of different structural types, such as theme identification and trend analysis [52–55]. In the era of big data, new ideas and methods have arisen in corpus linguistics research from natural language processing (NLP), as a branch of artificial intelligence [56]. The Python language is often used for text analyses in NLP, such as text processing and understanding, as well as semantic and sentiment analysis. It realizes data

analysis and processing by supporting various algorithms with overloaded operators and dynamic types for functions, modules, numbers, strings, and other elements [57].

As Chinese and English are structurally different, the word segmentation used in text mining for English texts is not suitable in the Chinese field [58]. Therefore, Chinese scholars have developed methods that are suitable for Chinese text mining in related data cleaning and analysis [59–61]. Among them, Hsu et al. discussed how text-mining techniques can be used to explore the features or knowledge contained in a large number of Chinese news documents and proposed a mining procedure suitable for Chinese news documents [58], as shown in Figure 2. It is carried out in two steps: the first step is to preprocess the text with existing and new words to extract key words; the second step is to set the mining mode and explore domain trends and feature differences in combination with domain knowledge. Based on this mining structure, this research mainly used the Python language to carry out three algorithms, namely, term frequency–inverse document frequency (TF-IDF), Simpson's diversity index (SDI) and latent Dirichlet allocation (LDA), as well as the text mining of the researched texts according to the three research purposes.

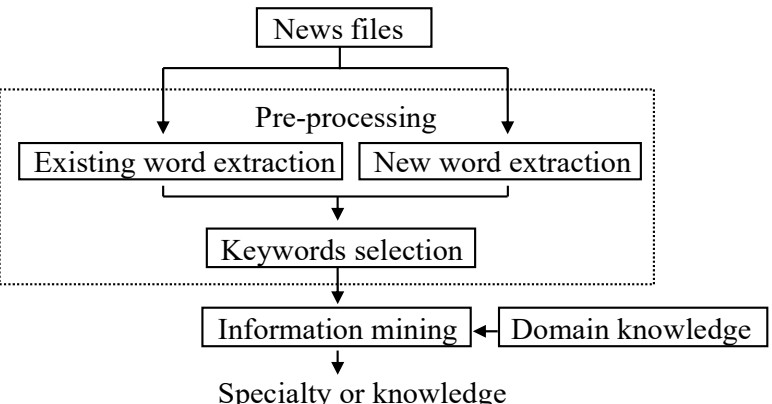

**Figure 2.** Chinese news document mining framework (Redraw form [58]. Copyright 2001 Hsu and Chen).

### 3.3. Research Process

Through the above-mentioned literature review, this study carried out empirical studies as follows (Figure 3): first, the key textual content of Industrial Design was preprocessed using Python and KJ statistics; then, the clustering analysis of the subjects was carried out using the K-means++ algorithm so as to study the evolution of theme, diversity, and sustainable design trends using TF-IDF, SDI, and LDA, respectively; finally, a conclusion was drawn.

### 3.4. Preprocessing of Data Text

Articles are mainly divided into three types: structured, semi-structured, and unstructured articles. Among them, a semi-structured news article usually briefly describes the news in its first paragraph, and therefore, important words in the text will appear in the first paragraph of the summary [58]. *Industrial Design* used semi-structured articles in the form of news reports from its first issues to issues 102, 104, 106, and 108, while issues 103, 105, 107, and 109–142 used structured articles in the form of academic papers. Based on these article characteristics and composition structures, this research finally obtained 1420 articles containing valid texts, including 948 semi-structured articles and 472 structured articles, by selecting the first paragraphs or the first 300 words of semi-structured articles of *Industrial Design*, as well as the titles, abstracts, and keywords of the structured articles from January 2008 to December 2020, excluding non-academic articles in the form of work displays, college introductions, overseas news, etc.

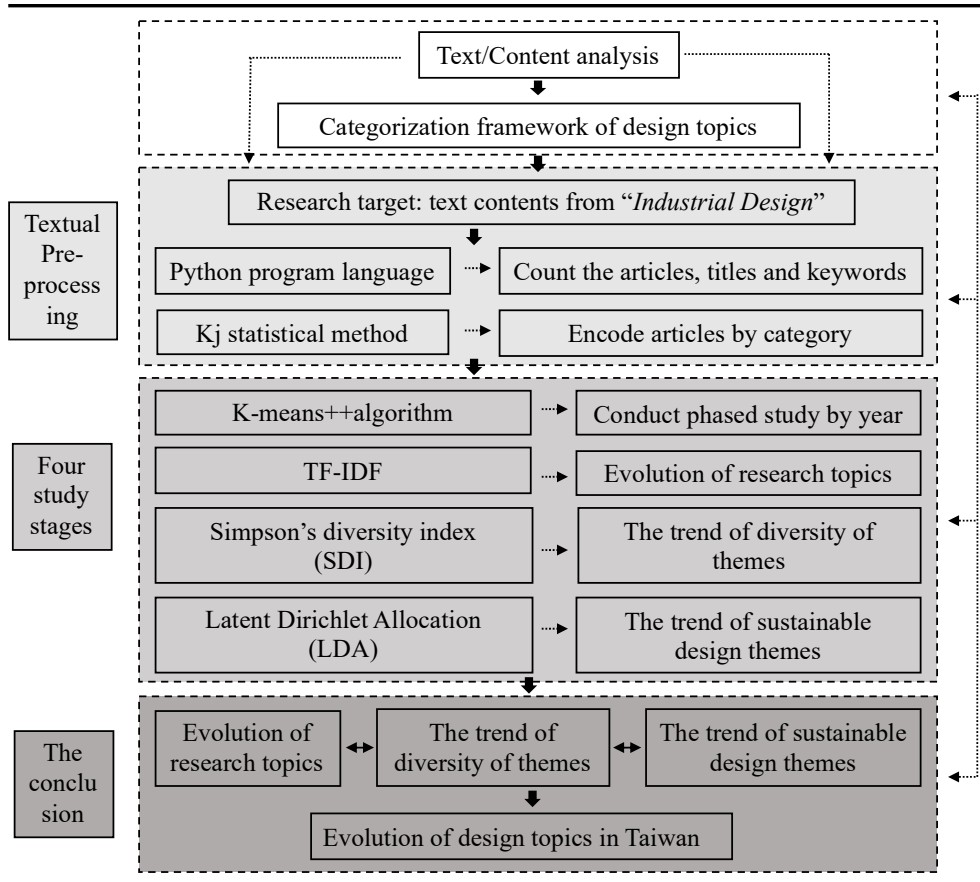

**Figure 3.** The flow chart.

Python was then uniformly used as the programming language. Regarding word segmentation, this research applied the jieba as its word-segmentation solution (http://github.com/fxsjy/jieba/ (accessed on 13 March 2021)), which is currently commonly used in academia. It was carried out in three steps: first, the keywords of each article were obtained in four processes, namely, eliminating single-character terms, extracting nouns and verbs, first paragraph vocabularies and frequency rules, and filtering general vocabularies; second, high-information vocabularies were further extracted using TF-IDF. This is a weighting method that considers the differences in text features. Compared with word frequency, TF-IDF accurately expresses the characteristics of the text [35]. Therein, stopwords were based on the Chinese stopwords database of Harbin Institute of Technology (https://github.com/goto456/stopwords/blob/master/hit_stopwords.txt (accessed on 13 March 2021), and further combined with manual input methods to add common vocabularies for design industry characteristics and to disable general vocabularies, such as design, influence, and other terms. Finally, for the vocabulary of the text research, the TF-IDF value in each article was calculated separately to select the top 20 representative words of the TF-IDF value as the research objects. After the above-mentioned text preprocessing, the basic statistics of the text data of *Industrial Design* are shown in Table 2.

**Table 2.** *Industrial Design* text data summary after text preprocessing (formulated in this research).

| Total Number of Articles | Total Number of Characters in Articles (Excluding Punctuation) | Total Number of Words in Articles | Total Number of Words after Removing Stopwords |
| --- | --- | --- | --- |
| 1420 | 273,003 | 121,400 | 58,439 |

### 3.5. Article Categorization and Coding

For the identification of key points of research topics in different articles, this research categorized 1420 *Industrial Design* articles to construct research topic categories. Analyses were conducted in two steps: In the first step, two design professional researchers conducted the coding work independently and categorized minor sub-topics against the main contents of each article by the KJ method. In the second step, minor sub-topics were categorized into middle sub-topics and then into major sub-topics by three experts based on the category list framework of Huang and Yan (2007) [30]. The specific category content of each sub-topic is shown in Appendix A. The basic data of the two researchers and three experts are shown in Table 3. The final categorization results are 273 minor sub-topics, 21 middle sub-topics, and 9 major sub-topics (Table 4).

**Table 3.** Basic Data of Interviewed Researchers and Experts.

| A | Researcher 1 | Researcher 2 | Expert 1 | Expert 2 | Expert 3 |
|---|---|---|---|---|---|
| B | PhD student, Creative Industry Design Institute, National Taiwan University of Arts | PhD student, Creative Industry Design Institute, National Taiwan University of Arts | Chairman of a design company, fashion brand president and brand director, PhD in design, regional visual director of *The Economist* magazine in Taiwan. | Master of Art Institute of National Central University, planning research of a design company | Visual director of a design company, assistant professor of visual design in a certain university. |
| C | Once | Once | Once | Once | Once |
| D | 2021.5.8 | 2021.5.8 | 2021.5.16 | 2020.5.27 | 2021.5.13 |

**Table 4.** Categorization of research topics of articles in *Industrial Design*.

| Major Sub-Topics | Middle Sub-Topics | Major Sub-Topics | Middle Sub-Topics |
|---|---|---|---|
| (BC1) Design communication and practical research | Visual practice design and application | (BC6) Design education | Design education development and practice |
| | Design culture research | | Design education research |
| (BC2) Design planning and execution | Design management and strategy | (BC7) Design theory | Basic theories and methods |
| | Design methods and procedures | | Design thinking and innovation |
| (BC3) Introduction to foreign design | Industry development | (BC8) Design technology | Theories and applications of intelligent technology |
| | Design education | | Space and planning design research |
| | Character event | | Digital media and design |
| (BC4) Design industry development research | Regional design research | (BC9) Perception and preference research | Principles and applications of human-factor engineering |
| | Industry development research | | Imagery and preference research |
| (BC5) Social service design | Service design issues | | |
| | Environmental and social issues | | |

In order to ensure the reliability of the categorization work, this study tested the reliability using inter-rater reliability and took the Cohen's kappa coefficients as a measure of reliability [62–64]. The 273 minor sub-topic categories were coded as SC1–SC273, the 21 middle sub-topic categories were coded as MC1–MC21, and the 9 major sub-topic categories were coded as BC1–BC9. The reliability of minor sub-topic category coding by the two researchers is shown in Table 5. The reliability of middle sub-topic category coding by the three experts is 0.813, and the reliability of major sub-topic category coding by the three experts is 0.511. All values are greater than 0.5, indicating that the reliability of the coding is within an acceptable range.

**Table 5.** Reliability of sub-topic category coding by two researchers.

| | SC1–14 | SC15–41 | SC42–57 | SC58–61 | SC62–85 | SC86–104 |
|---|---|---|---|---|---|---|
| Cohen's kappa | 0.954 | 0.674 | 0.861 | 0.556 | 0.541 | 0.774 |
| *p*-value | <0.000 | <0.000 | <0.000 | <0.000 | <0.000 | <0.000 |
| | SC105–118 | SC119–142 | SC143–150 | SC151–167 | SC168–181 | |
| Cohen's kappa | 0.649 | 0.566 | 0.809 | 0.669 | 0.876 | |
| *p*-value | <0.000 | <0.000 | <0.000 | <0.000 | <0.000 | |
| | SC182–200 | SC201–215 | SC216–225 | SC226–237 | SC238–242 | |
| Cohen's kappa | 0.508 | 0.619 | 0.502 | 0.543 | 0.500 | |
| *p*-value | <0.000 | <0.000 | <0.000 | <0.000 | <0.000 | |
| | SC243–249 | SC250–252 | SC253–260 | SC261–264 | SC265–273 | |
| Cohen's kappa | 0.744 | 0.947 | 0.855 | 0.686 | 0.757 | |
| *p*-value | <0.000 | <0.000 | <0.000 | <0.000 | <0.000 | |

## 4. Results

This section is divided into four parts: first, the division of the 60-year-long design development trend into years using the text K-means++ algorithm; second, the study of the evolution of theme throughout all periods; third, the study of the trend of diversity of themes throughout all periods via SDI; and fourth, the exploration of the sustainable design context of all periods using LDA.

### 4.1. Chronology of Design Research

It is necessary to divide the periods into stages so as to explore whether Taiwanese design research motifs show a noticeable difference through different periods. The encoded data first collected the keywords in articles by year and used the text K-means++ algorithm for clustering and visualized processing. Principal component analysis (PCA) was applied to reduce the dimensionality. The multi-dimensional data were converted into a low-dimensional space, so as to display the research subject in a more distinct way. Figure 4 shows the visualization of the clustering results.

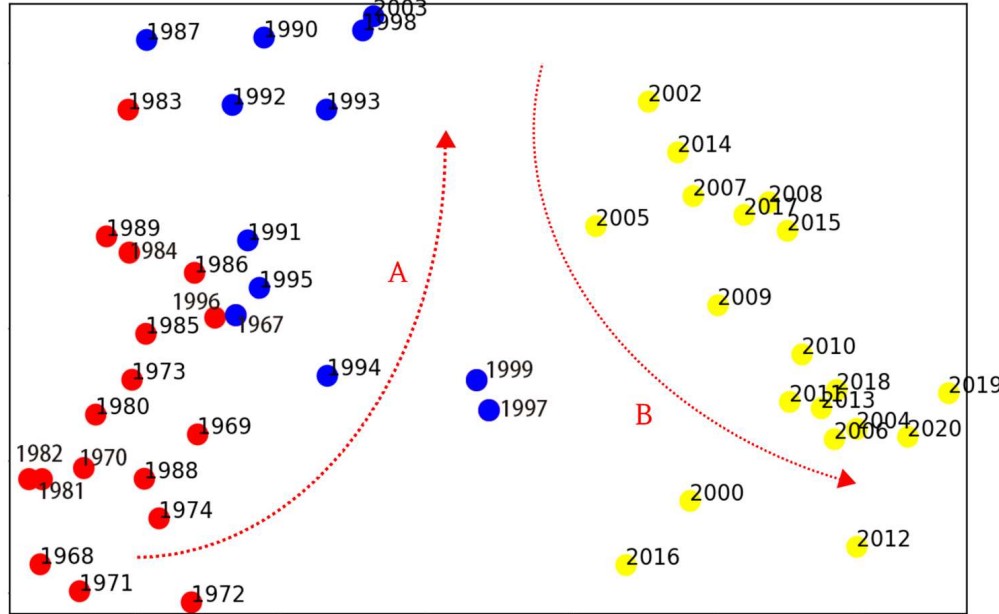

**Figure 4.** Matrix position of three clusters.

The results show that with the evolution of time, the chronological stage of the design field can be divided into three stages: in the first stage (1960s–1980s), the years corresponding to the red dots are y68–y90, y95, and y96; in the second stage (1990s), the years corresponding to the blue dots are y67, y91–y94, y97–y99, y03, and y05; in the third stage (2000s–2020), the years corresponding to the yellow dots are y00, y02, y04, and y06–y20. Among them, "y68" means 1968, "y91–y94" means 1991–1994, and so on, in a similar fashion.

### 4.2. Main Design Motifs at Each Stage

For the purpose of gaining a clearer understanding of the research topics and directions at the three stages, the categorization according to topic areas led to the results shown in Table 6, and Figure 5 shows the changes in the proportion of research on each topic. It can be seen from Table 6 that at the first stage (1960s–1980s), there was a total of 681 articles, of which "design planning and strategy" was the main topic (27.0%), followed by "introduction to foreign design" (20.6%). The third was "design theory" (18.0%). At the second stage (1990s), there was a total of 245 articles, of which "design theory" was the main topic (18.4%), followed by "design communication and practical research" (18.0%), and then "design technology" (13.5%). At the third stage (2000s–2020), there was a total of 494 articles; "perception and preference research" was the main topic at this stage (31.0%), followed by "design technology" (17.4%), and then "design planning and strategy" (16.6%).

**Table 6.** Main research topics at three stages.

|  | First Stage (1960s–1980s) | | Second Stage (1990s) | | Third Stage (2000s–2020) | |
|---|---|---|---|---|---|---|
| Design communication and practical research | 101 | 14.8% | 44 | 18.0% | 79 | 16.0% |
| Perception and preference research | 41 | 6.0% | 32 | 13.1% | 153 | 31.0% |
| Design technology | 45 | 6.6% | 33 | 13.5% | 86 | 17.4% |
| Design theory | 123 | 18.0% | 45 | 18.4% | 34 | 6.9% |
| Design education | 26 | 3.8% | 24 | 9.8% | 29 | 5.9% |
| Design planning and strategy | 184 | 27.0% | 31 | 12.7% | 82 | 16.6% |
| Social service design | 6 | 0.9% | 8 | 3.2% | 12 | 2.4% |
| Design industry development research | 15 | 2.2% | 6 | 2.4% | 16 | 3.2% |
| Introduction to foreign design | 140 | 20.6% | 22 | 9.0% | 3 | 0.6% |
| Total | 681 | 100% | 245 | 100% | 494 | 100% |

The above-mentioned data were made into a visual chart, as shown in Figure 5. The topics that continued to rise at the three stages are "perception and preference research", "design technology", and "design industry development research". The topic "introduction to foreign design" continued to decline. Education on design had a noticeable decline at the third stage.

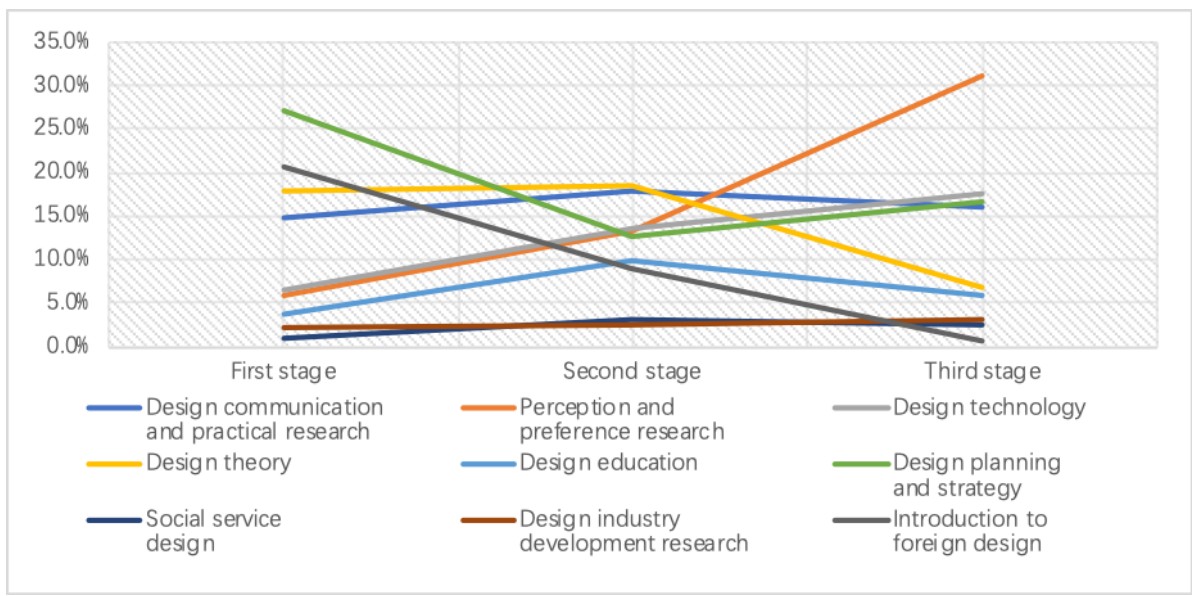

**Figure 5.** Changes in the proportions of research in various areas at three stages.

*4.3. The Study of the General Trend of the Design Context*

In order to analyze the overall trend of design motifs, the keywords for the above-mentioned three stages were sorted to select the top 20 terms based on the TF-IDF value (Figure 6). The TF-IDF values of keywords in text contents from *Industrial Design* at each stage were analyzed to detect the key vocabulary of each stage.

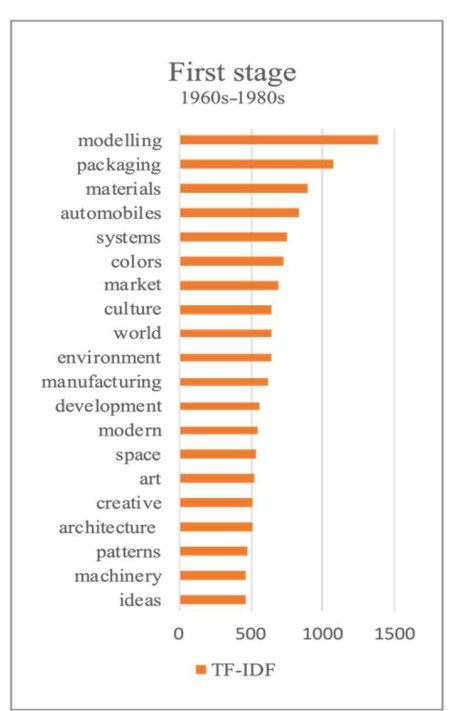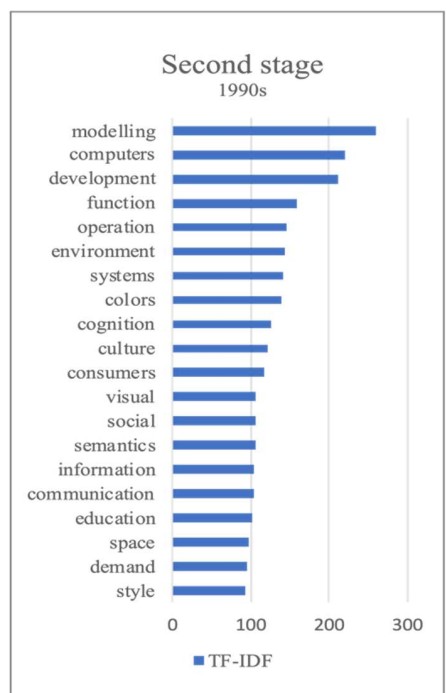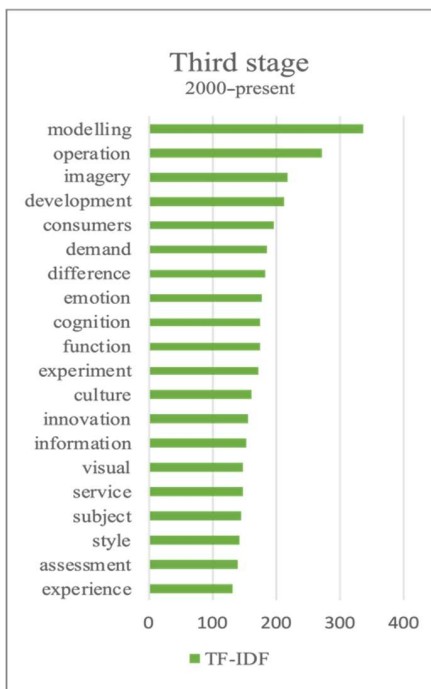

**Figure 6.** Keyword TF-IDF value ranking at three stages.

According to Figure 4, we may find two changing trends: Trend A (1967–2003) and Trend B (2002–2020). For a further analysis of original differences, a difference comparison was performed on research topics between the starting and ending groups of the two trends. The starting and ending groups of Trend A were defined as (y68, y7l, y72) and (y90, y98, y03), respectively, while the starting and ending groups of Trend B were (y02, y05, y07)

and (y18, y19, y20), respectively. We calculated the difference between the total number of papers in the ending group and the total number of papers in the starting group. The results are shown in Table 7. The larger the value of the difference in the table is, the greater the influence of this topic on the trend will be. The positive and negative numbers indicate whether the number of papers on the topic increased or decreased in the trend.

**Table 7.** Main evolutionary driving forces of Trends A and B.

| | | Trend A (1967–2003) | | | Trend B (2002–2020) | | |
|---|---|---|---|---|---|---|---|
| | | Research Topic | Category | | Research Topic | Category | |
| Based on downtrend | 1 | Design methods and procedures | Design planning and execution | −21 | Imagery and preference research | Perception and preference research | −6 |
| | 2 | Character event | Introduction to foreign design | −14 | Digital media and design | Design technology | −5 |
| | 3 | Basic theories and methods | Design theory | −13 | Design culture research | Design communication and practical research | −4 |
| Based on ascending difference | 1 | Principles and applications of human factors engineering | Perception and preference research | +5 | Regional design research | Design industry development | +2 |
| | 2 | Theories and applications of intelligent technology | Design technology | +1 | Theories and applications of intelligent technology | Design technology | +1 |
| | 3 | Design education research | Design education | +1 | Design education research | Design education | +1 |
| | | | | | Design methods and procedures | Design planning and execution category | +1 |

### 4.4. Diversity of the Study on Design

It is observed from the history of the study on design that design is an integrated application discipline, with numerous theories taken from other fields to explain issues in the field of design, which leads to a greater level of diversity of the study on design. Greater diversity boosts innovative study instead of restricting it to the existing frame. However, this extra diversity can lead to extra divergency and loss of focus regarding fields. As diversity can reflect the state of the fields themselves, the study of diversity bears significant value [65].

To explore the evolution trend of diversity of the previous studies on design, this study measured the diversity using quantized metrics, namely, SDI, which is extensively used in ecology. It can be used to measure the level of diversity in an enclosed system. It mainly utilizes pi, the proportion of the individual number of different categories in the system, to measure the diversity within the system with ($D = \sum pi^2$) and ($1/D$) [66]. The system's complexity and level of diversity grow as SDI increases. In this study, design is considered as a system. The main textual content is divided into nine major sub-topics to calculate pi for nine major sub-topics. The diversity index of design study topics for each year can be determined using SDI (Figure 7). In order to verify if it is a noticeable trend, this study aims to validate the integrating degree using regression analysis, with $R^2 = 0.375$ and sig = 0.008 ** ($p < 0.01$). It is statistically significant with respect to the trend.

### 4.5. The Sustainability of the Study on Design

In order to explore the sustainable theme in the literature of an unstructured enormous quantity, the LDA used in this study is an effective solution [67]. LDA serves as a non-supervision machine learning algorithm. As there is no need to label documents in advance, analysis can be performed relatively independently of a human's prior judgments. It has

been widely used as a technique for analyzing recent academic trends, as it is useful for finding hidden topics in the literature. Among several algorithms of topic modeling, LDA is widely used as a representative probabilistic topic model [68].

According to the broad concept of sustainable development and sustainable theme category [67,69,70], first, the analysis of frequent vocabularies was carried out on 1420 pieces of text using TF-IDF. The first 10% of vocabularies based on TF-IDF in each text were used to represent the key purport of that text. Thus, the themes with the first three weights of three stages, the first 10 words of each theme, and the proportion of that theme in the entire terminology database are to be exported by running LDA. However, the researchers should mark and name the theme and divide the research subject; the results are shown in Table 8.

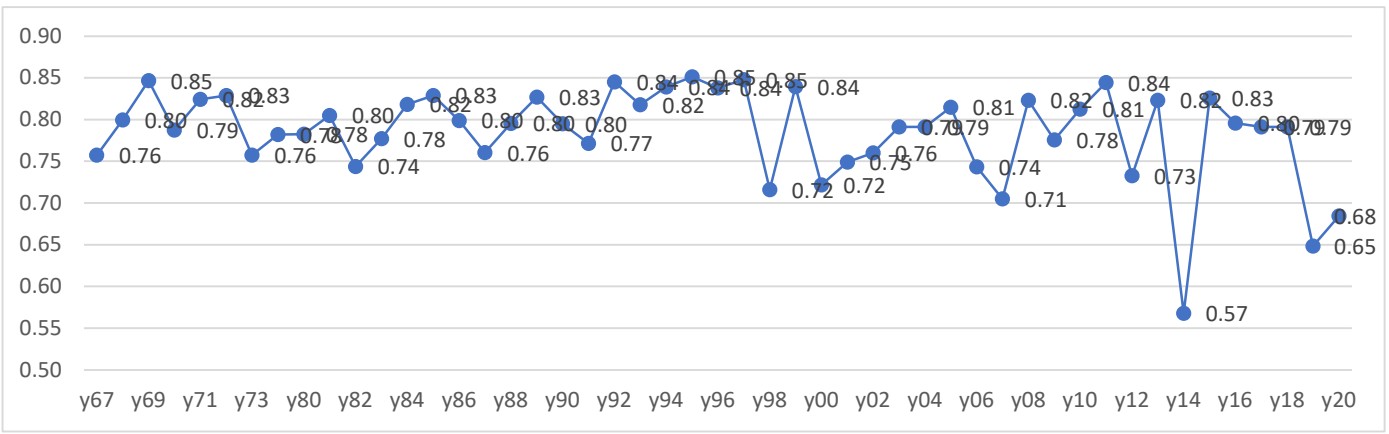

**Figure 7.** The evolution trend of the diversity index.

**Table 8.** Identified topics from the LDA.

| | Topic | Top 10 Relevant Words | Share |
|---|---|---|---|
| | | **The first stage (1960s–1980s)** | |
| 1 | IT and finance for sustainability | Packing, commodity, industry, Japan, appliance, lighting, automobile, material, market, printing | 0.022 |
| 2 | Sustainable marketing | Enterprises, model, symbols, issues, methods, society, cooperation, packing, commodity, activities | 0.021 |
| 3 | Education for sustainability | Education, plastics, space, students, buildings, life, craftsmanship, centers, materials, lectures | 0.020 |
| | | **The second stage (1990s)** | |
| 1 | Sustainable consumer behavior | Consumers, enterprises, transmission, positioning, products, special cases, technology, man-making, pictographic, bionic | 0.030 |
| 2 | Education for sustainability | Education, ecology, experts, basics, special topics, situation, products, style, lessons, users | 0.028 |
| 3 | Sustainable marketing | Users, selection, management, operation interfaces, toilets, recycling, quality, dismantling, waste, system | 0.028 |
| | | **The third stage (2000s–2020)** | |
| 1 | Sustainable community | Image, sensation, culture, evaluation, preference, personality, display, enterprises, model, picture books | 0.021 |
| 2 | Sustainable communication | Interface, users, operation, games, websites, ads, social community, function, experience, software | 0.021 |
| 3 | Sustainable development | Users, network, tribes, mobile phones, phubbers, sensation, the old, culture, factors, clients | 0.020 |

## 5. Discussion

In this section, the paper explores Taiwan's design history from the 1960s to 2020 based on the evolution of theme, diversity, and sustainability. It can be seen from the clustering results that the design topics in Taiwan can be divided into three stages, namely, the 1960s–1980s, the 1990s, and the 2000s to 2020.

### 5.1. The Analysis of Evolution of Design Themes

According to the changes in the proportion of research topics at each stage (Figure 5) and the keyword TF-IDF values (Figure 6), a dimensional construction was made for the major sub-topics, corresponding to the category contents of the middle and minor sub-topics of each major sub-topic (Appendix A), for the following analysis:

In terms of the key themes for each stage, first, in the 1960s–1980s, design motifs mainly included "design planning and strategy", "introduction to foreign design", and "design theory". It can be seen that the focus of design motifs was on the research and development of materials and technology, the popularization of basic theories, and the introduction of foreign design trends. Second, in the 1990s, the design motifs were mainly "design theory", "design communication and practical research", and "design technology". This shows that the focus of design motifs was on the interpretation of multiculturalism, the integration of digital design, and the communication of design aesthetics. Huang et al. also came to this conclusion in their research; they point out that that from 1996 to 2004, the most popular research topics in Taiwan's *Journal of Design* articles were design culture research, methods and strategies, and digital media and other themes [30]. Third, since the beginning of the 21st century, design motifs have been mainly categorized as "perception and preference research", "design technology", and "design planning and strategy", with technology integration and development in the design field; emotional human factors designs based on intelligent technology were the main issue of this period. Shih et al. also delivered the same results in their study [41].

In terms of upward and downward trends (Table 7), three topics that have continued to rise are "perception and preference research", "design technology", and "design industry development research", while the topic "introduction to foreign design" has continued to decline. According to the analysis, the middle and minor sub-topics that have continued to rise mainly include imagery and preference cognition, human factors analysis, scientific and technological theories and applications, industrial trends, localized design, and other related topics, which shows that the overall current design motifs are directed towards intelligent technology, emotional human factors, and localized culture as a whole, while the contents that have continued to decline include foreign design industry development, design education, and institutional competitions, which shows the importance of improving the integration of intelligent technologies, Kansei engineering, and other directions in the field of design education. However, these results are also consistent with those of Chao et al. [70].

In terms of the entire trend and driving forces, in Figure 5, it was found that there are two development trends: Trend A (1967–2003) and Trend B (2002–2020). In combination with Table 7 and middle sub-topics: in Trend A, "emotional human factors" and "digital technology" were the main driving forces for the evolution of design motifs from the 1960s to the 1990s, and in Trend B, "intelligent technology" and "local culture" were the main driving forces for the evolution of design motifs from the 2000s to 2020. Among the current driving forces, "intelligent technology" shows technology integrated with design across fields, and "local culture" is the competitiveness of the design culture. This trend also affirms the opinions of Shan et al. (2002) that "differentiation" based on "regionalization" has been the core of design development since the start of the 21st century [33]. These two directions are not only the main driving forces of the current trends of design motifs in Taiwan, but should also be considered to affect the future trends of design motifs. They can be regarded as a pilot balloon for design research.

*5.2. The Trend of Diversity Index of Design Themes*

It can be seen in Figure 7 that the rapid change in and diversity of study themes came as an anticipated, inevitable outcome of the influence of the beginning of, growth by, and frequent changes in design fields in the first stage (1960s–1980s); the index was on the rise as a whole during the second stage (1990s), which shows that design fields became wider with more diversified points of view; in the third stage (2000s–2020), the rising trend during the 1990s continued throughout the 2000–2006 period. Subsequently, acute fluctuations occurred. The analysis above, in combination with the topics mentioned, shows that more diversified points of view were born with VR and AI, and other technologies were incorporated into design after 2006. Thus, the diversity during this period rose through fluctuations. In addition, the index has been on a decline as a whole in recent years, which indicates that design may have become an independent and mature field, dedicated to a study theme such as smart technology or local culture.

It is certain that a higher level of diversity presents the existence of diversified points of view, which can constantly provide ways to think about issues from varied points of view [65]. In recent years, the declining diversity hints that current design needs to be stimulated by external environmental variations. All these are issues that should be discussed in subsequent studies.

*5.3. The Development Trend of Sustainable Design*

Since the mid-20th century, design has served as a major function of business innovation and been engaged in different aspects of sustainability discussions and practices in government and local communities [71–73]. Design has been recognized in the literature as a catalyst to move away from the traditional take–make–dispose model to achieve a more restorative, regenerative, and circular economy [74,75]; this study explored the trend of regional sustainable design by means of the development context of Taiwanese design.

As shown by the relevant vocabularies in Table 8, it was found that "IT and finance for sustainability", "sustainable marketing", and "education for sustainability" were comparatively important sustainable design directions during the 1960s–1980s. It can be seen that sustainable design was focused on technology, the market, and education at the beginning of the design era. This conclusion is also reflected in the global sustainability design trend in this period, and the relations among resource constraints, design technology, material production, and the environment are also among the key issues [76,77]. In the 1980s, the United Nations (UN) declared the importance of education for sustainable development and proposed a transition to green design and sustainable lifestyles [78–80].

As design moved towards a human-oriented nature in the 1990s, "sustainable consumer behavior" became important at this stage. However, compared to the last stage, "education for sustainability" was on the rise. The content of education shifted from the technology and methodology of the last stage to situation, style, and other design trends. Then, "sustainable marketing" moved to third place in order of importance. The vocabulary frequency shows that there was an awareness of sustainability at the stage of design in the market, such as the recycling, dismantling, and treatment of wastes. It can be seen from this stage that sustainable design was focused on consumers' behaviors and experience, as well as environmentally friendly design. During this period, global sustainability design also discussed the impact of the way consumers interact with products on the environment [15,81,82]; the proposal of green design, ecological design, and other approaches [83,84]; and enabling product personalization [85], designing products that age with dignity' [86], and other proposals of emotionally durable design [87]. It can be seen that during this period, Taiwan was relatively in line with the global sustainability design trend.

Since the beginning of the 21st century, "sustainable community"," sustainable communication", and "sustainable development" have become comparatively important sustainable design directions. With the interdisciplinary development of smart technology in design, it is often mentioned in terms of interfaces, social communities, preferences, etc. It

can be seen from this stage that integrating smart technology and local culture into Kansei engineering stands at the center of sustainable design. During this period, localization is also a hot topic for the global sustainable design issue. Cultural, social, and personal factors can have an important impact on design [88–90], as well as community development and enabling community members to express their opinions in design thinking [91] and explore sustainable concepts based on localized technology and recycled living in design education [92].

In sum, since the 1960s, sustainability design issues in Taiwan have gradually expanded from a focus on technology and products to large-scale system-level changes, such as consumer behavior, emotional design, and localization and technological innovation since the start of the 21st century. Such development is also consistent with the development of global sustainable design [87].

## 6. Conclusions and Suggestions

Compared with previous research into the main design topics, this paper attempts to categorize the development of modern design topics in Taiwan using a more complete journal text collection in order to explore the evolution of theme, diversity, and sustainable design trends in a historical context. It is shown by the clustering results that design topics in Taiwan can be divided into three stages, namely, the 1960s–1980s, the 1990s, and the 2000s–2020.

Regarding the evolution of theme, the research topics of these three stages became significantly different from one another over time. First, from approximately the 1960s to the 1980s, the design topics were focused on the research and development of materials and technologies, the popularization of basic theories, and the introduction of foreign design trends. Second, in the 1990s, the focus of design topics was on the interpretation of multiculturalism, the integration of digital design, and the communication of design aesthetics. Third, the focus of design topics in this period was on methods and procedures based on intelligent technology, emotional human factors, and local culture. The decline of design education indicates its potential to improve the integration of smart technologies and other hot topics across fields. Taiwan has shown two evolutionary trends since the development of modern design in the 1960s, namely, Trend A (1967–2003), which comprises "emotional human factors" and "digital technology", and Trend B (2002–2020), which comprises "local culture" and "intelligent technology".

Regarding diversity, design themes were subject to rapid changes and a higher level of diversity at the beginning of the design era during the 1960s–1980s. In the 1990s, design themes were on the rise as a whole in terms of diversity. This shows that design has been incorporated into Kansei engineering, digital technology, and other fields; from 2000 to 2020, the diversity of design themes was subject to rapid changes with the rise of smart technology, local culture, and other topics. In recent years, diversity has begun to drop, indicating that current design requires stimulation through external environmental variations, which warrants further investigation in subsequent research.

Regarding the trend of sustainable design, it was focused on technology, the market, and education from approximately the 1960s to the 1980s; it was focused on human factors and experience design centered on consumers in the 1990s, with design education's direction centering on situation and style. Furthermore, emphasis was placed on eco-design, as well as the integration of smart technology, local culture, and Kansei engineering across fields during the 2000–2020 period.

This study makes the following contributions: For industrial applications, it can help companies understand the current design situation and predict future trends; from a theoretical perspective, the results of this research can help academic researchers understand the research trends shown in the design field and then determine current and future potential research directions; it can also aid researchers in exploring further possibilities of research methodologies in regional design and supplementing the Chinese design context aim using the diversity of global design history; in the context of sustainability, such exploration

will not only help to establish the history of regional sustainable design, but also explore the development direction of sustainable and mutualistic growth of design and regional resources based on the historical context. For the purpose of future research recommendations, horizontal comparisons may be made between the design development contexts in Taiwan and other regions or countries to enable researchers to investigate the differences between Taiwan and the international design context, which contributes to analyzing the development of design topics in regions from a multi-dimensional perspective.

**Author Contributions:** Each author contributed to the paper. Conceptualization, J.H.; data curation, Y.T.; formal analysis, Y.T.; methodology, J.H.; software, M.S.; visualization, M.S.; writing—original draft, J.H.; writing—review and editing, M.S. and J.W. All authors have read and agreed to the published version of the manuscript.

**Funding:** The authors gratefully acknowledge the support for this research provided by the 2022 Fujian Social Science Foundation Project, NO. FJ2022BF055; and The 2022 "Fourteenth Five" Plan for Educational Science in Fujian Province, NO. FJJKBK22-054.

**Institutional Review Board Statement:** Not applicable.

**Informed Consent Statement:** Not applicable.

**Acknowledgments:** The author hereby expresses thanks to the teachers—in particular, the original idea came from Rungtai Lin and Po-Hsien Lin's directing—and students of the Creative Industry Design Institute of National Taiwan University of the Arts for their help, and to the interviewed experts in academic and industry circles for their support of this research. The author also expresses thanks for the funding from MOST—108-2221-E-144-001. Special thanks to the review committee for giving suggestions to improve this paper.

**Conflicts of Interest:** The authors declare no conflict of interest.

## Appendix A

**Table A1.** List of Major, Middle, and Minor Sub-topics Involved in Articles of *Industrial Design*.

| Research Topic Category | | Research Area |
|---|---|---|
| **Major Sub-Topics** | **Middle Sub-Topics** | **Minor Sub-Topics** |
| BC1:<br>Design planning<br>and execution | MC1:<br>Design methods and procedures | SC1–14: Material and technology, product design case introduction, photography technology, lighting design, toy design, illustration design, information technology, automotive design, laser technology, electromechanical system, product development, innovative technology, styling design, modular design |
| | MC2:<br>Design management<br>and strategy | SC15–41: Design strategy, design procedure, design management, intellectual property, design evaluation, market analysis, design application, development and cost, project analysis, design evaluation, design performance, design contract, design patent, business merger, system design method, product cycle, review system, design standards, market pricing, design department, product evaluation model, diversified marketing, product planning, design resources, design communication, corporate social responsibility, interview skills |

**Table A1.** *Cont.*

| Research Topic Category | | Research Area |
|---|---|---|
| **Major Sub-Topics** | **Middle Sub-Topics** | **Minor Sub-Topics** |
| BC2:<br>Design communication and practical research | MC3:<br>Design culture research | SC42–57: Color physical research, crafts, craft seminars, design style, design history, theme connotation, modeling art, aesthetics, multiculturalism, regional cultural differences, regional design history, model meaning and essence, ethnic culture, local characteristics of culture industry, rationality and sensibility, design aesthetics |
| | MC4: Issues for disadvantaged groups | SC58–61: Issues such as aging, people with disabilities, children, women, etc. |
| | MC5:<br>Visual practice design and application | SC62–85: Visual communication, advertising, text and layout design, symbols, images, illustrations, fonts, visual design case introduction, packaging design, ID case analysis, cross-domain visual design, pointer design, corporate identification system, CI strategy, composition, pattern, expression technique, icon, CIS design, parameterization, cross-domain academic theory, perspective drawing method, brand image, brand awareness |
| BC3:<br>Perception and preference research | MC6:<br>Imagery and preference research | SC86–104: Modeling psychology, visual psychology, color perception, product image, preference, host and guest psychology, design semantics, consumer demand, usage context, consumer life style, demand pattern, style image, institutional experience, pleasant design, user evaluation, consumer experience, emotional design, interactive design, experience design |
| | MC7:<br>Principles and applications of human factors engineering | SC105–118: Engineering concepts, cognitive research, identification, behavioral science, anatomy, physiology, cognitive psychology, human factors engineering, human factors analysis, Kansei engineering, visual cognition, auditory cognition, creative cognition, perceptual image |
| BC4:<br>Design theory | MC8:<br>Design thinking and innovation | SC119–142: Design conception, design philosophy, creativity, visual communication thinking, designer interviews, design experience, design trends, differences between east and west ideas, design performance, automotive industry, design concepts, role concepts, design concepts, design inspiration, design Analysis, design method, design language, theoretical model, design meaning, design literacy, serialized design, universal design, Zen idea, integrated design |
| | MC9:<br>Basic theories and methods | SC143–150: Basic design theory, popularization of design concepts, design research methods, design guidelines, questionnaires, design plans, cross-domain theory, academic architecture |
| BC5:<br>Design technology | MC10:<br>Theories and applications of intelligent technology | SC151–167: Neural network, virtual technology, human-machine environment system, projection system, robot system, information processing system, video on-demand system, electric bicycle design, cross-domain design, virtual design, portable electronic products, intelligent recommendation system, eyeball tracking, forward-looking technology, artificial intelligence, animation design, platform development |
| | MC11:<br>Digital media and design | SC168–181: Web design, digital issues, Internet, automation, software applications, interface design, digital design method, computer-aided design, digital history, multimedia tools, new media art, digital development trends, 3D animation, information design |
| | MC12:<br>Space and planning design | SC182–200: Space design concept, thinking, architectural space design, spanning design, furniture design, living environment, exhibition design, exposition, support organization, transportation space design, display system, three-dimensional, future theme, three-dimensional modeling, three-dimensional image, online games, urban street scenes, spatial landscapes |

**Table A1.** *Cont.*

| Research Topic Category | | Research Area |
|---|---|---|
| **Major Sub-Topics** | **Middle Sub-Topics** | **Minor Sub-Topics** |
| BC6:<br>Design education | MC13:<br>Design education research | SC201–215: Design education seminars, educational thinking, new ideas, theme design, industry-university cooperation, educational methods, school-running models, educational concepts, educational analysis, design educational history, teaching principles and methods, educational management and innovation, design teams, evaluation system, cross-domain cooperation |
| | MC14:<br>Design education development and practice | SC216–225: Curriculum research, graduation production exhibition, school status, design teaching method, design competition, teaching achievement exhibition, subject theory, Internet, software teaching, practical teaching |
| BC7:<br>Social service design | MC15:<br>Environmental and social issues | SC226–237: Bionic design concept, environmental design, natural energy, power source, ecological analysis, green design, sustainable issues, environmentally friendly materials, green design seminars, environmental product development, green design evaluation, sustainable design |
| | MC16: Service design issues | SC238–242: Designer rights, customer participation in design, consumer participation in design, service design, co-design |
| BC8:<br>Design industry development research | MC17:<br>Industry development research | SC243–249: Industrial development trends, design associations, design exhibitions, design competitions, seminars, product design weeks, design agencies |
| | MC18: Regional design research | SC250–252: Localization issues, community design, local resources |
| BC9:<br>Introduction to foreign design | MC19: Industry trends | SC253–260: Industrial development, case analysis, product introduction, design agency, design concept, design policy, design industry development history, design yearbook |
| | MC20: Design education | SC261–264: Foreign design education, curriculum analysis, design school, craft education |
| | MC21: Character event | SC265–273: Designers, design groups, fairs, international design associations, design competitions, design sports, foundations, works exhibitions, travel notes |

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
