# Peer review of "Empirical Study on Design Trend of Taiwan (1960s–2020): The Evolution of Theme, Diversity and Sustainability"

_sustainability, doi:10.3390/su141912578_

Round 1
Reviewer 1 Report
1.The manuscript is well written. The purpose, methods and results are clearly presented.
2.Huang and Yan’s(2007) paper is more important for this study, it should be described more and compare the result in discussion section.
3.Please clarify the Python algorithm in the abstract.
4.Please check, Line 326: Table7 should be Table 6; Line 473: Figure 8 should be Figure 7; Line 500: Table 9 should be Table 8.
Author Response
Point 1: Huang and Yan’s(2007) paper is more important for this study, it should be described more and compare the result in discussion section.
Response 1:There is the addition of the comparison of and the discussion over the Huang and Yan’s(2007) study results in section 5.1. See L390-L393 for the revisions.
Point 2: Please clarify the Python algorithm in the abstract.
Response 2: The detailed Python algorithm has been added into the abstract. See L15- L18.
Point 3: Please check, Line 326: Table7 should be Table 6; Line 473: Figure 8 should be Figure 7; Line 500: Table 9 should be Table 8.
Response 3: These errors have been corrected accordingly, see L296, L426, L451.

Reviewer 2 Report
The topic of the manuscript is quite interesting and valuable, and I am pleased to recommend it to publish in the journal after an appropriate revision. When I carefully read the manuscript, I found that many texts were repetitive and imprecise, and there was relatively little discussion in sustainability, which may be conflict to the journal scope. Therefore, it is suggested that the length of the article can be shortened, and the irrelevant contents can be removed (at least, simplified) as much as possible. In addition, many minor errors were found throughout the manuscript, which largely reduced the readability and quality of the article. Thus, the authors should reconfirm and correct these inappropriate parts in detail.
Many capitalization problems, inconsistences, wrong formats, as well as typos, such as in lines of 20, 21, 44, 45, 48, 55, 61, 83, 93, 124, 129, 134, 138, 145-146, 177, 209, 263, 277, 389, 414, 443, 467… (not for all), and the author likes to use the term “scholar(s)” (more than 11 times), it is recommended to be modified appropriately.
L71, When saying “The design of Asia has become a key role.” Do the authors have any evidence (such as references), or this is a fact that is well recognized by all designers worldwide?
L89, Abbreviations should be defined at first mention and used consistently thereafter, i.e., OEM, ODM and OBM. Please check this point throughout the manuscript.
Please note the paper being submitted to the journal Sustainability. When reporting “Furthermore, as Taiwan is typically one of resource-limited areas, the scholars note that exploring Taiwan’s sustainable development trend bears indicative significance in globalization [25, 26, 27].” I believe that this is one of the most crucial points for motivating the study. Unfortunately, the references cited here were out of date (1996-2005). More newly published papers, as well as more information, should be mentioned to strengthen the study rationale and motivation.
L124, design researches? A strange term and please check it.
L121-125, This research has conducted a systematic study in the historical context of Taiwan’s design topics and analyzed the current design trends with a view to providing scholars with references for design researches in terms of academic researching; in terms of practical application, helping companies understand current design development trends.
I suggest that the authors should highlight the role of sustainability to match the journal scope more.
In L115, the authors have highlighted the three main study objectives, and in the next paragraph the authors mentioned about “The purpose of a historical research is to provide reference value…”, the contents seemed to be redundant. This part should be modified to more concise descriptions to avoid confusing.
In 142-147, the authors mentioned again that “…this paper aims to organize the development context of modern design in Taiwan in different directions from the 1960s to the present on four aspects including design policy…”, please merge these contents into the study objective in the end of Introduction section.
Figure 1 seems to be developed previously by Lin MT and Lin RT [23], please consider the copyright issue.
L192, 3.2 Research Methods, in this subsection, I suggest the authors should focus on how (not why) the study did, even though in general the authors are allowed to mention some reasons why they adopted the methods and analyzing tools to make readers more clearly understand the study protocols.
L196, The purpose of what?
L209, was this statement cited by the reference #55?
L219, Figure 2, again, this figure was developed by a previous study.
Compared to the Introduction and Methods sections, the results were very specific and valuable. However, I still notice that the narratives of the manuscript are still a bit disorganized and redundant and there should be also a distinction between the results obtained in the study and that from the past studies, which should be clearly stated to avoid confusion. Finally, it is recommended to strengthen the discussion of sustainability, as this is a key reason for the journal to accept this paper for publication.
L576, is any ethical code available?
Author Response
Point 4: The length of the article can be shortened, and the irrelevant contents can be removed (at least, simplified) as much as possible. In addition, many minor errors were found throughout the manuscript, which largely reduced the readability and quality of the article. Thus, the authors should reconfirm and correct these inappropriate parts in detail.
Response 4: Those repeated and rough content has been deleted and revised extensively. And the minor errors have been reconfirmed and corrected.
Point 5: L71, When saying “The design of Asia has become a key role.” Do the authors have any evidence (such as references), or this is a fact that is well recognized by all designers worldwide?
Response 5: For this question, the relevant literature has been added. See L75.
Point 6: L89, Abbreviations should be defined at first mention and used consistently thereafter, i.e., OEM, ODM and OBM. Please check this point throughout the manuscript.
Response 6: The issues herein and these hereinafter have been revised.
Point 7: Please note the paper being submitted to the journal Sustainability. When reporting “Furthermore, as Taiwan is typically one of resource-limited areas, the scholars note that exploring Taiwan’s sustainable development trend bears indicative significance in globalization [25, 26, 27].” I believe that this is one of the most crucial points for motivating the study. Unfortunately, the references cited here were out of date (1996-2005). More newly published papers, as well as more information, should be mentioned to strengthen the study rationale and motivation.
Response 7: For this opinion, recently-published literature has been quoted . See L103.
Point 8: L124, design researches? A strange term and please check it.
Response 8: This phrase has been revised. The paragraph with a description of the contribution from the researches has been merged into the discussion in article 6 to avoid repeated description. See L525--L534.
Point 9: The authors should highlight the role of sustainability to match the journal scope more in L121-125.
Response 9: For the contribution from the researches, the contributions regarding sustainable design has been added herein. See L531 - L534.
Point 10: In L115, the authors have highlighted the three main study objectives, and in the next paragraph the authors mentioned about “The purpose of a historical research is to provide reference value...”, the contents seemed to be redundant. This part should be modified to more concise descriptions to avoid confusing.
Response 10: This part has been deleted and revised. This paragraph has been changed into one for the description of the overall structure of the paper. See L120 - L125.
Point 11: In 142-147, the authors mentioned again that “...this paper aims to organize the development context of modern design in Taiwan in different directions from the 1960s to the present on four aspects including design policy...”, please merge these contents into the study objective in the end of Introduction section.
Response 11: This part has been revised, see L135-137.
Point 12: Figure 1 seems to be developed previously by Lin MT and Lin RT [23], please consider the copyright issue. L219, Figure 2, again, this figure was developed by a previous study.
Response 12: The source of Figure 1 and 2 has been noted, See L172, L203.
Point 13: L192, 3.2 Research Methods, in this subsection, I suggest the authors should focus on how (not why) the study did, even though in general the authors are allowed to mention some reasons why they adopted the methods and analyzing tools to make readers more clearly understand the study protocols.
Response 13: This section has been revised by downsizing why to select and adding how a research method should be carried out. See L182-L187 and L194-L201 for the revision.
Point 14: L196, The purpose of what?
Response 14: The narration has been revised. See L177 - L178.
Point 15: L209, was this statement cited by the reference #55?
Response 15: Where such word has been quoted has been identified in the note. See L192-L194.
Point 15: The narratives of the manuscript are still a bit disorganized and redundant and there should be also a distinction between the results obtained in the study and that from the past studies, which should be clearly stated to avoid confusion.
Response 15: In section 4.4 and 4.5, the current results and the previous ones have been reorganized and clarified.
Point 16: Finally, it is recommended to strengthen the discussion of sustainability, as this is a key reason for the journal to accept this paper for publication.
Response 16: In section 5.3, the description of the research trend regarding global sustainable design through all three stages has been added, with a comparison made between Taiwan’s sustainable design context and the global one during the same period. See L453 - L502 for the revisions. See L444-L490.
Point 17: L576, is any ethical code available?
Response 17: As it is an empirical study on published papers herein, none of informed consent statement has been involved. So the description has been revised. See L546.

Reviewer 3 Report
It is a valuable attempt to analyze the design trend of Taiwan from an empirical study by using Text Mining and Python. The discussion of previous research is representative of the existing knowledge and conclusions are well connected with the results discussion. However, the structure of the paper is a bit confused. It could help clarify the paper if at the end of the introduction, in addition to concisely explaining the methodology, the organization of the different sections is explained and justified. On the other hand, some issues are redundant; e.g., too many references are made to the objectives and methodology throughout the text. There are also problems with the syntax rules, punctuation marks and capitalization among others.
Author Response
Point 18: The structure of the paper is a bit confused. It could help clarify the paper if at the end of the introduction, in addition to concisely explaining the methodology, the organization of the different sections is explained and justified.
Response 18: The description of this paper’s organizational structure has been given to the last paragraph in ‘Introduction’ so that the readers can have a clearer understanding of the paper’s structure. See L120-125.
Point 19: Some issues are redundant; e.g., too many references are made to the objectives and methodology throughout the text.
Response 19: The repeated and extra content herein has been revised or downsized extensively.
Point 20: There are also problems with the syntax rules, punctuation marks and capitalization among others.
Response 20: The entire paper has been corrected in terms of proper use of language.

Round 2
Reviewer 2 Report
I really appreciate the authors for their revision works after I carefully read the responses to comments and compare the differences between the original and revised manuscript. The current version of manuscript has been much improved, particularly the research methods. The revisions for the issues are reasonable and acceptable. I have no further comment about the manuscript.